

# Genotype-specific responses to *in vitro* drought stress in myrtle (*Myrtus communis* L.): integrating machine learning techniques

Ümit Bektaş[1], Musab A. Isak[2], Taner Bozkurt[3], Dicle Dönmez[4], Tolga İzgü[5], Mehmet Tütüncü[6] and Özhan Simsek[1,2]

[1] Faculty of Agriculture, Department of Horticulture, Erciyes University, Kayseri, Turkey
[2] Graduate School of Natural and Applied Sciences, Agricultural Sciences and Technologies Department, Erciyes University, Kayseri, Turkey
[3] Tekfen Agricultural Research Production and Marketing Inc., Adana, Turkey
[4] Biotechnology Research and Application Center, Çukurova University, Adana, Turkey
[5] Institute of BioEconomy, National Research Council of Italy, Florence, Italy
[6] Department of Horticulture, Ondokuz Mayis University Samsun, Samsun, Turkey

Corresponding author
Özhan Simsek, ozhan12@gmail.com

## ABSTRACT

**Background:** Myrtle (*Myrtus communis* L.), native to the Mediterranean region of Türkiye, is a valuable plant with applications in traditional medicine, pharmaceuticals, and culinary practices. Understanding how myrtle responds to water stress is essential for sustainable cultivation as climate change exacerbates drought conditions.

**Methods:** This study investigated the performance of selected myrtle genotypes under *in vitro* drought stress by employing tissue culture techniques, rooting trials, and acclimatization processes. Genotypes were tested under varying polyethylene glycol (PEG) concentrations (1%, 2%, 4%, and 6%). Machine learning (ML) algorithms, including Gaussian process (GP), support vector machine (SVM), Random Forest (RF), and Extreme Gradient Boosting (XGBoost), were utilized to model and predict micropropagation and rooting efficiency.

**Results:** The research revealed a genotype-dependent response to drought stress. Black-fruited genotypes exhibited higher micropropagation rates compared to white-fruited ones under stress conditions. The application of ML models successfully predicted micropropagation and rooting efficiency, providing insights into genotype performance.

**Conclusions:** The findings suggest that selecting drought-tolerant genotypes is crucial for enhancing myrtle cultivation. The results underscore the importance of genotype selection and optimization of cultivation practices to address climate change impacts. Future research should explore the molecular mechanisms of stress responses to refine breeding strategies and improve resilience in myrtle and similar economically important crops.

## INTRODUCTION

Myrtle, also known as *Myrtus communis* L., is a perennial evergreen shrub or small tree native to the Mediterranean region (*Cioć et al., 2018*). It belongs to the Myrtaceae family, which consists of 100 genera and 3,000 species that grow naturally in tropical and subtropical regions (*Rezaee & Kamali, 2014*; *Şimşek et al., 2023*). Various parts of the myrtle plant, including its leaves, flowers, and fruits, contain numerous important constituents for the medical, food, liquor, and cosmetic industries (*Serce et al., 2008*). These constituents make myrtle useful for anti-genotoxic, anti-mutagenic, antiseptic, anti-inflammatory, and treating internal and tropical infections (*Bonjar, 2004*). Essential oils derived from myrtle can be utilized in pharmaceutical production, and their leaves are used in fragrance and food industries (*Mulas & Cani, 1999*). Myrtles can also be consumed as a tea (*Flamini et al., 2004*; *Şimşek et al., 2022*). This shrub or small tree typically ranges in height from 0.5 to 3 m, and two subspecies have been identified: *M. communis* subsp. communis and *M. communis* subsp. tarentina (*Picci & Atzei, 1996*).

Myrtles can thrive in various habitats and are consistently the dominant plant in the maquis. They are slow-growing, long-lived evergreen plants that do not shed their leaves during winter (*Sumbul et al., 2011*). The leaves of the myrtle plant have an aromatic odor, and the fruits are berry-like, typically blackish-purple or white, and ripen during the autumn season (October–December). The ripe fruits are both sugary and astringent and are pollinated by insects. The fruits are covered with a light, waxy layer that helps bees recognize the plant, giving the fruit a matte appearance. The fruits are multi-seeded and have a pleasant, spicy, and aromatic odor. They contain essential oils, tannins, sugars, and organic acids (citric and malic acids) (*Aleksic & Knezevic, 2014*). The leaves of the myrtle plant also contain small amounts of phenolic acids such as caffeic, ellagic, gallic acid, and quercetin derivatives, as well as large amounts of catechin and myricetin derivatives (*Simsek et al., 2020*).

Drought is a significant contributor to the decline in plant production. Environmental stress hinders the growth and development of plants by limiting their ability to respond and adapt to their surroundings. The high biological complexity involved in a plant's response to stress is often accompanied by numerous changes at various levels, including morphological, molecular, cellular, physiological, and biochemical. *M. communis* has demonstrated remarkable resistance to abiotic stressors; however, the prolonged drought conditions and intense solar radiation experienced during Mediterranean summers may have contributed to its decline (*Gucci et al., 1997*; *Mendes, Gazarini & Rodrigues, 2001*).

Plants cope with drought stress by regulating their metabolism through osmosis. This process involves the active accumulation of organic and inorganic compounds, known as osmolytes or osmotic protectors, which occur with decreased cell water potential. The process of osmotic regulation in cells allows osmotic potential to be reduced with the help of osmotic protectors that accumulate at high rates (*Sanchez et al., 2004*).

Several studies have found that testing plant responses to abiotic stress factors under *in vitro* conditions is a dependable method that produces consistent results compared to field experiments. This approach allows for the evaluation of many genotypes under stress

conditions in a practical manner (*Basu, Gangopadhyay & Mukherjee, 2002*). For instance, an *in vitro* study (*Di Cori et al., 2013*) examined the adaptation mechanisms of myrtle to salinity stress. However, the physiological processes through which *M. communis* responds to water stress or drought have not been thoroughly documented (*Gratani, Catoni & Varone, 2013*; *Navarro et al., 2009*; *Tafreshi et al., 2021*).

In recent years, *in vitro* studies have used polyethylene glycol (PEG), sorbitol, and mannitol as abiotic stress factors to create stress conditions, drought agents, pathogens, or phytotoxins directly in the pathogen culture medium as abiotic stress agent (*Manoj, Chaudhary & Singh, 2021*). In tissue culture, PEG, sorbitol, and mannitol were employed as osmotic stress agents. While PEG is commonly used to create artificial drought conditions by reducing the available water potential in the culture media, it is preferred because it does not cause phytotoxic effects in the culture media (*Hassan et al., 2004*).

Machine learning (ML) applies data science techniques to address complex challenges across various scientific domains (*Bozkurt et al., 2024*; *Tarraf et al., 2024*). Artificial intelligence-based ML models can analyze and predict complex, multivariate datasets (*Kul et al., 2020*; *Isak et al., 2024*; *Tütüncü et al., 2024*) and learn intricate relationships (*Jamshidi et al., 2020*; *Sadat-Hosseini et al., 2022*) to overcome these issues. These techniques have gained popularity in recent years for analyzing datasets in multiple fields within plant science. However, machine learning methodologies in plant and agricultural sciences are relatively limited compared to their extensive application in other scientific disciplines (*Aasim et al., 2023*; *Şimşek et al., 2024a*). Nevertheless, some researchers have achieved notable success in various areas of plant science, such as plant breeding (*van Dijk et al., 2021*) and gene function (*Mahood, Kruse & Moghe, 2020*). Adopting ML techniques offers the advantage of enabling computers to learn independently and convert data into valuable knowledge without human programming (*Hesami et al., 2022*). Table 1 presents the history of the related literature review.

This study explores the drought stress responses of different myrtle genotypes, hypothesizing that these genotypes display varying levels of tolerance to drought conditions. Myrtle, a highly valued Mediterranean plant with diverse applications, is increasingly confronted with drought due to global climate change. In this regard, the study employs ML analysis utilizing the Gaussian process (GP), support vector machine (SVM), Random Forest (RF), and Extreme Gradient Boosting algorithms (XGBoost) to model and predict micropropagation and rooting efficiency across varying concentrations of polyethylene glycol.

To test this hypothesis, the study aims to: (1) utilize morphological assessments to identify changes related to drought stress; (2) employ ML models to predict drought tolerance based on the observed and morphological data.

## MATERIALS AND METHODS

### Plant material
In this study, four distinct myrtle genotypes were selected through a meticulous process conducted in Karaisalı, Adana, and Erdemli, Mersin, Türkiye, in 2017. The selection process involved the initial screening of 100 plants across various locations to identify the
**Table 1 Literature review analysis.**

| Species | *In vitro* method | ML model | Main findings | References |
|---|---|---|---|---|
| *Myrtus communis L.* | Cadmium stress | MLP, RF, XGBoost | White-fruited genotype showed greater resilience to cadmium stress. Machine learning models accurately predicted plant responses to environmental stress. | *Tütüncü et al. (2024)* |
| *Myrtus communis L.* | Solid media propagation and plant bio reactor | – | Plantform bioreactor system successful in myrtle micropropagation and rooting. | *Şimşek et al. (2023)* |
| *Myrtus communis L.* | Drought stress | – | Evaluation of PEG-induced water stress on regenerated shoots | *Tafreshi et al. (2021)* |
| *Myrtus communis L.* | *In vitro* shoot multiplication | | Selected genotypes cultured *in vitro* showed variation in multiplication rate. | *Ruffoni, Mascarello & Savona (2010)* |
| *Chrysanthemum* | *In vitro* sterilization | MLP, GA | Results indicated that the differences between the MLP predicted, and validation data were negligible | *Hesami, Naderi & Tohidfar (2019)* |
| *Petunia hybrida* | *In vitro* sterilization | GRNN, MLP, and RBF | The results showed that NSGA-II effectively optimizes disinfectant levels and immersion time to minimize contamination and maximize germination. | *Rezaei et al. (2023)* |
| *Lycium barbarum L.* | *In vitro* cadmium streaa | MLP | MLP and RF models accurately predicted plant physiological changes due to Cd exposure, with $R^2$ values up to 0.98. | *Isak et al. (2024)* |
| *Nicotiana tabacum* | Agrobacterium-mediated gene transformation | MLP | Sensitivity analysis of ANN models showed that the Agrobacterium strain was the most influential parameter in tobacco transformation. | *Niedbała, Niazian & Sabbatini (2021)* |
| *Passiflora caerulea* | Indirect shoot regeneration | GRNN, RF | Both RF and GRNN algorithms showed high predictive accuracy ($R^2 > 0.86$) in training and testing sets for all studied parameters. | *Jafari & Daneshvar (2023)* |
| *Passiflora caerulea* | *In vitro* rooting | GRNN, GA | The GRNN-GA model effectively predicts and optimizes *in vitro* rooting of *P. caerulea*, demonstrating both reliability and accuracy. | *Jafari et al. (2022)* |
| *Lavandula L.* | *In vitro* micropropagation | MLP, RBF, XGBoost, and GP | Machine learning models enhance efficiency and provide valuable insights. | *Şimşek et al. (2024a)* |
| Strawberry | *In vitro* drought | MLP, SVM, RF and GP | PEG concentrations affected plant height and multiplication coefficients. | *Şimşek (2024)* |
| *Cicer arietinum L.* | *In vitro* regeneration | MLP, SVR GP, XGB and RF | RF algorithm displayed the best performance to predict the outputs. | *Kirtis, Aasim & Katırcı (2022)* |

most promising genotypes. Specific quantitative criteria were used, including fruit yield measurements, fruit size assessments, and evaluations of plant architecture for robustness and resilience. The genotypes were categorized based on these parameters, focusing on identifying those with superior traits. Ultimately, two black-fruited and two white-fruited genotypes were selected, each demonstrating high performance in the critical visual parameters outlined. For the study, four genotypes were subjected to four different concentration levels, with ten plants per concentration and two replications for each genotype.

## Tissue culture studies

### Sterilization of shoot tips

The plant material's shoot tips were sterilized before being placed into the culture. To achieve this, the shoot tips transported to the laboratory were thoroughly washed under tap water for 10 min. In sterilization, the shoot tips were immersed in a 70% ethanol solution for 3 min, followed by a 10-min immersion in a 20% sodium hypochlorite solution. Lastly, the shoot tips were thoroughly rinsed three times using sterile distilled water within a sterile hood to eliminate any remaining sterilizing agents.

### Culturing and micropropagation of plants

Following the sterilization, shoot tips from the plant material were subsequently cultured on MS nutrient media containing varying concentrations of BAP (0, 1, and 2 mg/L) and agar (8 g/L). The plants were subcultured every 4 weeks, with three subcultures. The cells were cultivated at 25 °C under a 16-h light and 8-h darkness regime. PEG6000 was added to the medium to induce drought stress at concentrations of 0% as control, 2%, 4%, and 6% (*Tafreshi et al., 2021*). Future studies will explore higher concentrations to assess severe drought responses.

### Establishment of rooting trail

To evaluate the efficacy of IBA in promoting root growth, rooting trials were conducted using solid MS nutrient media containing varying concentrations of IBA (0, 1, and 2 mg/L) and 8 g/L agar. The plants were cultivated under controlled conditions of 25 °C and a 16-h light/8-h darkness cycle for 6 weeks. PEG6000 was added to the medium at four different concentrations (0% as control, 2%, 4%, and 6%) to simulate drought stress.

### Acclimatization

Upon completing the rooting trials, the plants were subjected to acclimation to external conditions. This involved gradually opening the culture containers before transferring the rooted plants to vials containing a 1:1 mixture of sterile peat and perlite in a greenhouse setting. The plants were removed from the agar medium using water.

## Data analysis

All trials were established utilizing a factorial arrangement within a completely randomized design, comprising four replicates, each consisting of five plants per replicate. In the micropropagation experiments carried out under *in vitro* conditions, plants were subcultured every 4 weeks, with three subcultures. The collected data underwent an analysis of variance (ANOVA). *Post hoc* pairwise comparisons of means for treatments deemed significant by ANOVA were conducted using the LSD test. Statistical analyses were executed utilizing the R programming language.

## Modeling procedure

This study employed a comprehensive suite of ML models to analyze the drought stress responses of various myrtle genotypes. The models used included GP, SVM, RF, XGBoost, and artificial neural network (ANN) based MLP to model and predict the

micropropagation and rooting efficiency of various myrtle genotypes using polyethylene glycol (PEG). Each model was chosen for its specific strengths in handling different data types and predictive tasks. For instance, GP and SVM are known for their robustness in classification and regression tasks, particularly with small datasets. At the same time, RF and XGBoost are powerful ensemble methods that are effective in handling large datasets and reducing overfitting. MLP, an ANN model, is particularly useful for capturing complex non-linear relationships in the data.

We utilized a 10-fold cross-validation method to divide the dataset into training and testing subsets and thoroughly evaluate the predictive performance of the ML models. The four genotypes served as input variables, with an additional concentration of PEG (0%, 2%, 4%, 6%) functioning as an input variable. On the other hand, micropropagation rate, plant height, plant height in the rooting medium, root number, and root length were the target (output) variables. We utilized R programming with the help of the Caret and Kernlab packages to implement the coding. Several metrics were used to assess and compare the accuracy and precision of the ML models, including the coefficient of determination ($R^2$), which indicates the degree of relationship between the model and the dependent variable Eq. (1), root mean square error (RMSE) which typically shows the precision of the model Eq. (2), and the mean absolute error (MAE), which calculates the average error between the predicted and observed values Eq. (3).

$$R^2 = 1 - \frac{\sum_{i=1}^{n} \left(Y_i - \hat{Y}_i\right)^2}{\sum_{i=1}^{n} \left(Y_i - \tilde{Y}\right)^2} \tag{1}$$

$$RMSE = \sqrt{\frac{\left(\sum_{i=1}^{n} \left(Y_i - \hat{Y}_i\right)^2\right)}{n}} \tag{2}$$

$$MAE = \frac{1}{n}\sum_{i=1}^{n} \left|Y_i - \hat{Y}_i\right| \tag{3}$$

**Multilayer perceptron**

The MLP is a commonly used example of an ANN, characterized by its input layer, output layer, and one or more hidden layers. To train the MLP, a supervised training approach was employed, utilizing the input and output variables from the training set. The training procedure was repeated until the desired value specified in Eq. (4) was achieved (*Şimşek, 2024*).

$$E = \frac{1}{n}\sum_{n=1}^{n} (y_s - \hat{y}_s) \tag{4}$$

## Gaussian process

To gain a deeper understanding of the propagation of random variables, the GP model for supervised learning extends the Gaussian probability distribution. This renders the GP model more suitable for addressing issues related to classification and regression. By determining the likelihood that input samples will fall within specific classes, it functions as a non-parametric classifier for binary datasets. One of the main advantages of this model is its ability to perform well with small datasets, as it provides consistency, accuracy, and ease of computation (*Hussein, El-Kerdany & Afifi, 2016*). The derivation process for each input (x) and its corresponding output (y) is detailed in Eq. (5).

$$y_i = f(x_i) + \varepsilon \tag{5}$$

## Support vector machines

SVMs are a class of artificial intelligence models that encompass supervised and unsupervised learning techniques. These models are particularly well-suited for tasks such as regression analysis, clustering, and classification. This section introduces three variations of SVMs: support vector classification (SVC), support vector regression (SVR), and two separate SVM variations. One of the key advantages of SVMs is their efficiency, even with relatively small datasets. This is in contrast to many other artificial intelligence algorithms, which often require large amounts of training data for optimal performance. Additionally, SVMs are less prone to problems such as overfitting, low convergence rates, and getting stuck in local minima, which are commonly associated with traditional artificial intelligence methods (*Aasim et al., 2022a*). The SVM algorithm, as described by Eq. (6), helps to determine which class has the furthest separator plane.

$$f(x) = w\varphi(x) + b \tag{6}$$

## Random forest

The RF ensemble learning method is an ensemble of unpruned trees that has been successful in both regression and classification tasks. It is well-known for its efficiency and ease of design, and previous research has identified several prominent characteristics of the RF model, including its ability to avoid overfitting, proficient noise handling, and efficient feature management (*Hu et al., 2019*).

To minimize the correlation between distinct trees while maintaining their unique strengths, two random sources are injected into each tree during construction for classification accuracy. Each tree is first trained by randomly selecting a replacement (bootstrap replica) for the training set. The algorithm then considers a small variable subset that is randomly chosen from the complete variable set to find the optimal split at each node. Additionally, every tree is fully grown, producing a low bias and significant variance in the tree outputs (*Breiman, 2001*; *Hu et al., 2019*). Regression models were solved using the Mean Squared Error (MSE) metric, which was used to determine the
distance between nodes and best branching choices within the forest. The fundamental idea is illustrated in Eq. (7).

$$y \; = \; \sum_{i=1}^{n} \left(a_i - a_i^*\right) k(x, x_i) + b \tag{7}$$

## Extreme gradient boosting (XGBoost) model

The XGBoost Model, developed by *Chen & Guestrin (2016)*, is a powerful tool for addressing regression and classification challenges. As a member of the gradient boosting decision tree family, XGBoost is renowned for its exceptional performance and speed. By functioning within a gradient boosting framework, XGBoost is particularly adept at learning from errors and progressively reducing the error rate over multiple rounds.

$$y_i = F(x_i) = \sum_{(d=1)}^{D} f_d(x_i), \; f_d \in F, \; i = 1, \; \dots, n \tag{8}$$

$$L_j = \sum_{i=1}^{n} l(y_{i,} \quad \hat{y}^{(j-1)} + \; f_j(x_i) + \Omega\left(f_j\right) \tag{9}$$

The XGBoost iterative model is expressed in Eq. (9), while its objective function is provided by Eq. (8).

# RESULTS

## Micropropagation

Four different genotypes of myrtle (two black-fruited and two white-fruited) were cultured on their shoot tips on a solid medium supplemented with 1 mg/L BAP. Different concentrations of PEG (0, 2, 4, and 6 mg/L) were added to these media to simulate drought stress. Table 2 provides data on micropropagation rate, showcasing the impact of varying concentrations of PEG) (0%, 2%, 4%, and 6% on the average growth of four distinct genotypes: White Fruited-1, White Fruited-2, Black Fruited-1, and Black Fruited-2. Notably, there was an observable pattern of diminishing growth as PEG concentration increased. For instance, the White Fruited-1 and White Fruited-2 genotypes declined average growth from 10.06 to 4.97 and from 10.00 to 5.03, respectively, as PEG concentration increased. Similarly, the Black Fruited-1 and Black Fruited-2 genotypes also decreased average growth with increasing PEG concentration.

By evaluating these values, it became evident that the highest average growth was consistently observed at 0% PEG for all genotypes. On the other hand, the lowest average growth was consistently observed at the highest PEG concentration of 6% for each genotype. This further supports the trend of decreasing growth as PEG concentration increases.

Table 3 displays the Plant Length data, demonstrating the impact of various PEG concentrations (0%, 2%, 4%, and 6%) on the plant length of genotypes White Fruited-1, White Fruited-2, Black Fruited-1, and Black Fruited-2. The average plant lengths for each

**Table 2 The effect of genotype and PEG concentrations on micropropagation rate.**

|  | PEG-0% | PEG-2% | PEG-4% | PEG-6% | Genotype average |
|---|---|---|---|---|---|
| White Fruited-1 | 10.06 | 9.30 | 6.97 | 4.97 | 7.825B |
| White Fruited-2 | 10.00 | 8.83 | 7.43 | 5.03 | 7.825B |
| Black Fruited-1 | 10.97 | 10.40 | 8.30 | 6.13 | 8.950A |
| Black Fruited-2 | 10.90 | 9.73 | 8.60 | 6.67 | 8.975A |
| PEG average | 10.48A | 9.57B | 7.82C | 5.70D |  |

**Table 3 The effect of genotype and PEG concentrations on plant length.**

|  | PEG-0% | PEG-2% | PEG-4% | PEG-6% | Genotype average |
|---|---|---|---|---|---|
| White Fruited-1 | 4.29 | 3.76 | 3.28 | 2.95 | 3.57B |
| White Fruited-2 | 4.31 | 3.81 | 3.22 | 2.98 | 3.58B |
| Black Fruited-1 | 4.71 | 3.99 | 3.53 | 3.19 | 3.87A |
| Black Fruited-2 | 4.48 | 4.11 | 3.64 | 3.24 | 3.86A |
| PEG average | 4.45A | 3.92B | 3.42C | 3.09D |  |

genotype at different PEG levels are provided, along with the Genotype Average (Table 2). Generally, the highest plant lengths across all genotypes were observed in the black-fruited genotype in the absence of PEG (0%) at a rate of 4.71, while the lowest plant lengths were recorded in the White Fruited-1 genotype at the highest PEG concentration (6%) at a rate of 2.95.

Table 4 also displays data on Plant Length in Rooting Medium, showcasing the influence of various PEG concentrations (0%, 2%, 4%, and 6%) on the plant length of White Fruited-1, White Fruited-2, Black Fruited-1, and Black Fruited-2 genotypes. The average plant length for each genotype at different PEG levels was calculated, resulting in the Genotype Average. The data presented in Table 3 indicates that the highest plant lengths in the rooting medium were observed in Black Fruited-1, with no PEG present (0%), at a rate of 4.17, across all genotypes. Conversely, the shortest plant length in the rooting medium was recorded in White Fruited-2 at the highest PEG concentration (6%), at a rate of 2.52.

## In vitro rooting

This study also analyzed the root number of four distinct genotypes under various concentrations of PEG, as detailed in Table 5. Notably, Black-Fruited-1 showed the highest root number (4.10) at 6% PEG, while Black-Fruited-2 had the lowest root number (2.03) without PEG. The data revealed a specific response of each genotype to PEG, with a noticeable trend of decreasing root numbers as PEG concentrations increased. This inverse relationship implies that higher PEG concentrations may inhibit root development across the examined genotypes, offering valuable insights into the intricate interactions between environmental factors and root growth.

 

**Table 4 The effect of genotype and PEG concentrations on plant length in rooting medium.**

|  | PEG-0% | PEG-2% | PEG-4% | PEG-6% | Genotype average |
|---|---|---|---|---|---|
| White Fruited-1 | 3.97 | 3.53 | 2.98 | 2.63 | 3.27B |
| White Fruited-2 | 3.76 | 3.43 | 2.83 | 2.52 | 3.13A |
| Black Fruited-1 | 4.17 | 3.52 | 3.08 | 2.79 | 3.39A |
| Black Fruited-2 | 3.73 | 3.54 | 3.22 | 2.57 | 3.36B |
| PEG average | 3.90A | 3.50B | 3.02C | 2.63D | |

**Table 5 The effect of genotype and PEG concentrations on number of roots.**

|  | PEG-0% | PEG-2% | PEG-4% | PEG-6% | Genotype average |
|---|---|---|---|---|---|
| White Fruited-1 | 2.47 | 3.37 | 3.77 | 3.90 | 3.37A |
| White Fruited-2 | 2.77 | 3.06 | 3.53 | 3.60 | 3.24A |
| Black Fruited-1 | 2.40 | 3.16 | 3.50 | 4.10 | 3.29A |
| Black Fruited-2 | 2.03 | 3.00 | 3.27 | 3.60 | 2.97B |
| PEG average | 2.41D | 3.15C | 3.51B | 3.80A | |

**Table 6 The effect of genotype and PEG concentrations on root length.**

|  | PEG-0% | PEG-2% | PEG-4% | PEG-6% | Genotype average |
|---|---|---|---|---|---|
| White Fruited-1 | 2.75ij | 2.94gh | 3.09efg | 3.27de | 3.01B |
| White Fruited-2 | 2.55k | 2.76hij | 2.97g | 3.50bc | 2.95B |
| Black Fruited-1 | 2.93ghi | 3.07fg | 3.23def | 3.52b | 3.19A |
| Black Fruited-2 | 2.64jk | 2.94gh | 3.34cd | 3.77a | 3.17A |
| PEG average | 2.72D | 2.93C | 3.16B | 3.51A | |

Note:
Capital letters indicate statistical differences between myrtle varieties and different PEG concentrations, while small letters indicate statistical differences between myrtle varieties with different PEG concentrations as determined by the LSD analysis.

Table 6 reveals the impact of four distinct myrtle genotypes and four varying concentrations of PEG on root length. The results show that Black Fruited-2 displayed the highest root length of 3.77 at a 6% PEG concentration, while White Fruited-2 exhibited the lowest root length of 2.55 without PEG. Additionally, there is a discernible trend of decreasing root numbers as PEG concentrations increase.

## Machine learning analysis

Table 7 provides a comprehensive evaluation of various ML models in different plant growth prediction tasks and allows for identifying the models with the highest performance metrics. Among these models, determining the models with the highest $R^2$ scores and lowest MAE values is crucial in determining the best performer regarding predictive accuracy and precision. Regarding the Micropropagation Rate, the MLP model emerged as the top performer with the highest $R^2$ score of 0.58, indicative of its superior predictive capability. Additionally, boasting the lowest MAE of 0.11 further solidifies the

**Table 7 Assessment metrics for the machine learning models.**

|  | ML models | $R^2$ | MAE | RMSE |
|---|---|---|---|---|
| Micropropagation rate | MLP | 0.58 | 0.11 | 0.14 |
|  | GP | 0.56 | 0.10 | 0.13 |
|  | SVM | 0.55 | 0.11 | 0.14 |
|  | RF | 0.54 | 0.11 | 0.14 |
|  | XGBoost | 0.52 | 0.12 | 0.15 |
| Plant height | MLP | 0.80 | 0.06 | 0.08 |
|  | GP | 0.79 | 0.07 | 0.09 |
|  | SVM | 0.78 | 0.06 | 0.08 |
|  | RF | 0.78 | 0.07 | 0.09 |
|  | XGBoost | 0.80 | 0.06 | 0.09 |
| Plant height in rooting medium | MLP | 0.79 | 0.08 | 0.10 |
|  | GP | 0.78 | 0.08 | 0.10 |
|  | SVM | 0.77 | 0.08 | 0.10 |
|  | RF | 0.77 | 0.08 | 0.10 |
|  | XGBoost | 0.78 | 0.08 | 0.10 |
| Number of roots | MLP | 0.32 | 0.14 | 0.17 |
|  | GP | 0.29 | 0.15 | 0.19 |
|  | SVM | 0.28 | 0.14 | 0.17 |
|  | RF | 0.27 | 0.15 | 0.18 |
|  | XGBoost | 0.24 | 0.17 | 0.20 |
| Root length | MLP | 0.46 | 0.13 | 0.16 |
|  | GP | 0.41 | 0.14 | 0.17 |
|  | SVM | 0.40 | 0.15 | 0.18 |
|  | RF | 0.39 | 0.15 | 0.19 |
|  | XGBoost | 0.50 | 0.12 | 0.15 |

**Note:**
ML, Machine Learning; MLP, Multilayer Perceptron; GP, Gaussian Process; SVM, Support Vector Machines; RF, Random Forest; $R^2$, Coefficient of Determination; MAE, Mean Absolute Error; RMSE, Root Mean Square Error.

MLP's position as the optimal choice for predicting the micropropagation rate. The ranking of the models in predicting the micropropagation rate is as follows: MLP < GP < SVM < RF < XGBoost. For the prediction of Plant Height, both MLP and XGBoost exhibited equally impressive $R^2$ scores of 0.80, suggesting a comparable predictive probability. However, considering the MAE values, MLP outperformed XGBoost with a lower value of 0.06, cementing its superiority in accurately predicting plant height. MLP and XGBoost emerged as the top performers, followed by GP, SVM, and RF, which exhibited comparable performances (MLP < XGBoost < GP = SVM = RF). Similarly, in the task of predicting Plant Height in Rooting Medium, MLP and XGBoost once again shared the highest $R^2$ score of 0.79. However, with a lower MAE of 0.08 compared to XGBoost's 0.09, MLP takes the lead as the preferred model for this specific prediction task. MLP and XGBoost maintained their superiority,

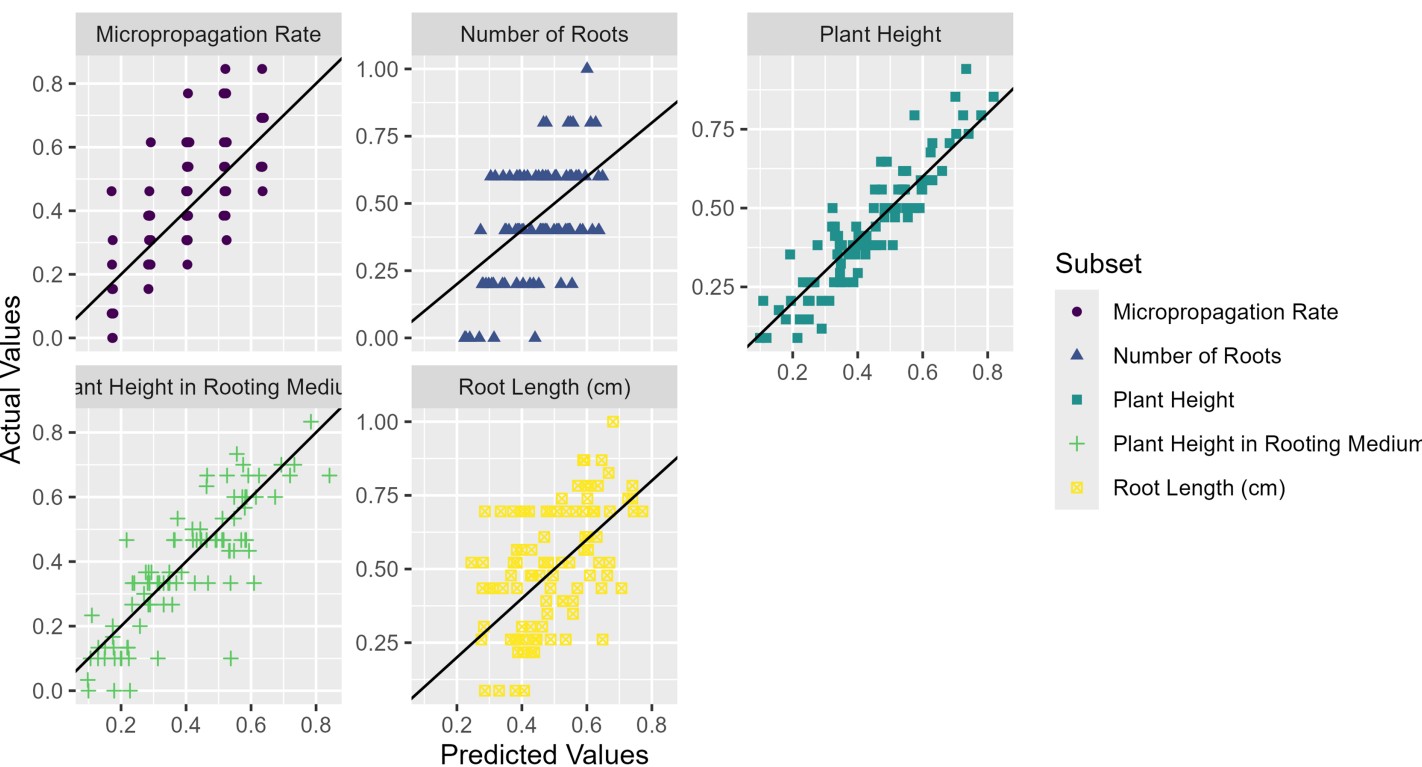

**Figure 1  Scatter plot actual against predicted values using GP model analysis.**

followed by GP, SVM, and RF, which demonstrated similar performance levels (MLP < XGBoost < GP = SVM = RF).

When it comes to predicting the Number of Roots, the multilayer perceptron (MLP) model attained the highest $R^2$ score of 0.32. However, its performance in terms of mean absolute error (MAE) was inferior, with a value of 0.14. Conversely, the SVM model demonstrated superior accuracy in predicting the number of roots, with a slightly lower $R^2$ score of 0.28 but a lower MAE of 0.14, positioning it as the preferred model. The SVM model emerged as the top choice, followed by MLP, RF, GB, and XGBoost (SVM < MLP < RF < GB < XGBoost). For predicting Root Length, the XGBoost model exhibited the highest $R^2$ score of 0.50 and the lowest MAE of 0.12, showcasing its exceptional performance in accurately forecasting root length. XGBoost was the most effective model, followed by MLP, with GB, SVM, and RF exhibiting comparable performance levels (XGBoost < MLP < GB = SVM = RF).

In conclusion, while the MLP model consistently achieved high $R^2$ scores across various tasks, its performance in terms of MAE varied. Therefore, it is crucial to consider both the $R^2$ scores and MAE values when determining the optimal model for each specific plant growth metric. The actual and predicted values are illustrated in Figs. 1–5, which compares the samples on the horizontal axis with the model's predicted outcomes on the vertical axis.

## Actual and Predicted Value of RF Model

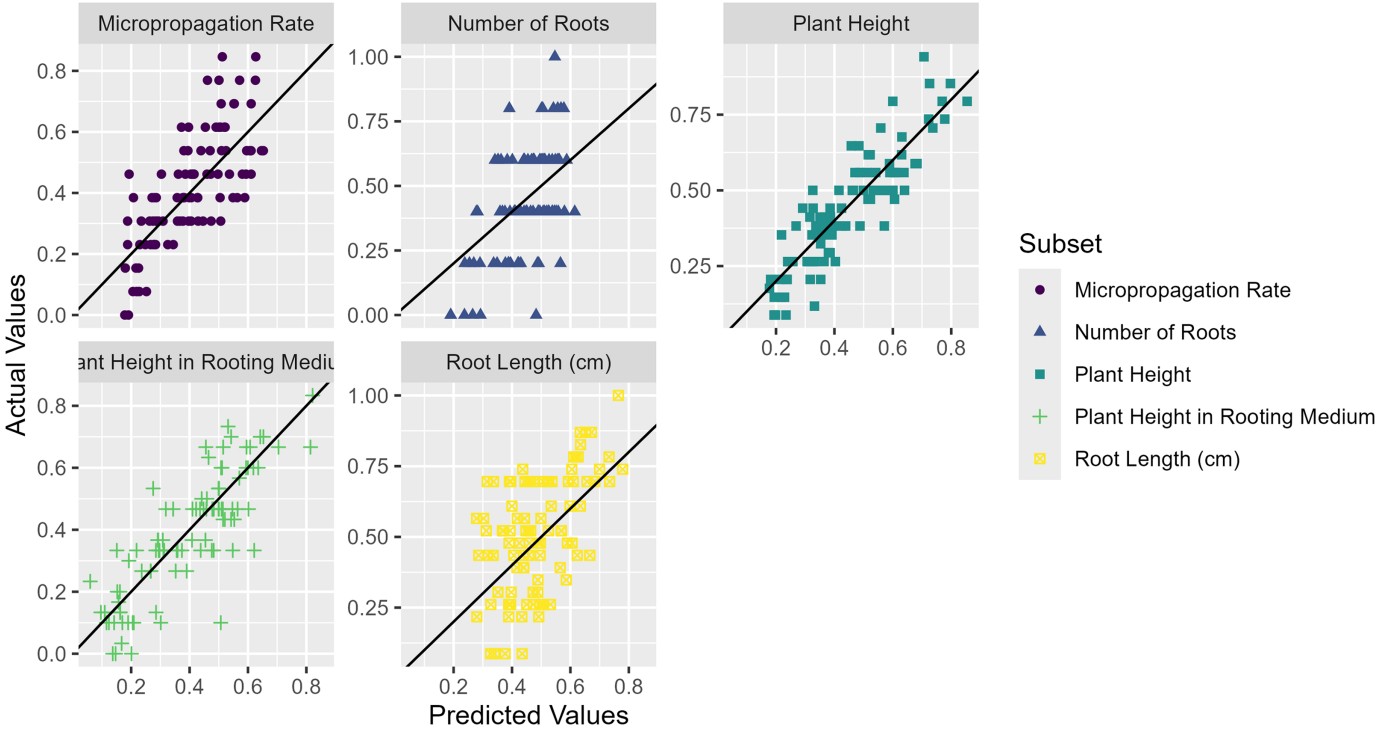

**Figure 2** Scatter plot actual against predicted values using RF model analysis. 

## Actual and Predicted Value of SVM Model

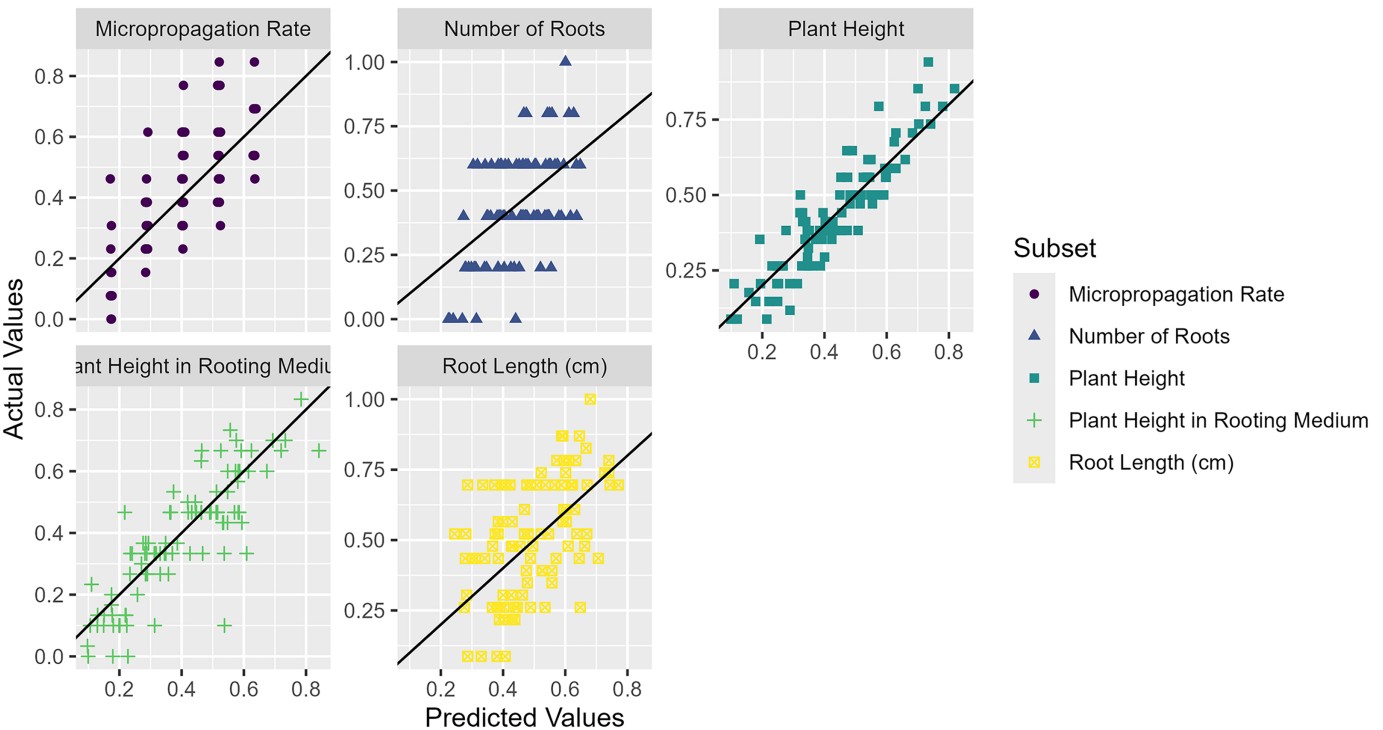

**Figure 3** Scatter plot actual against predicted values using SVM model analysis. 
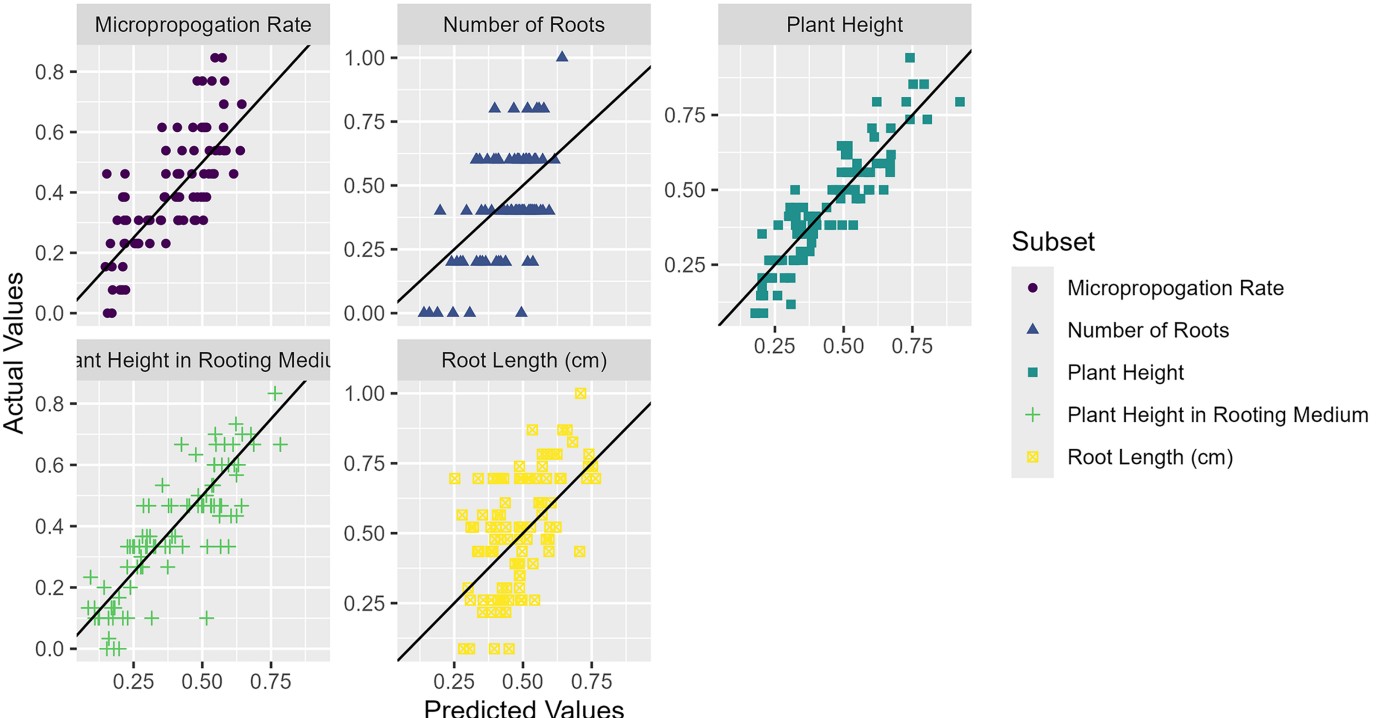

**Figure 4 Scatter plot actual against predicted values using XGBoost model analysis.**

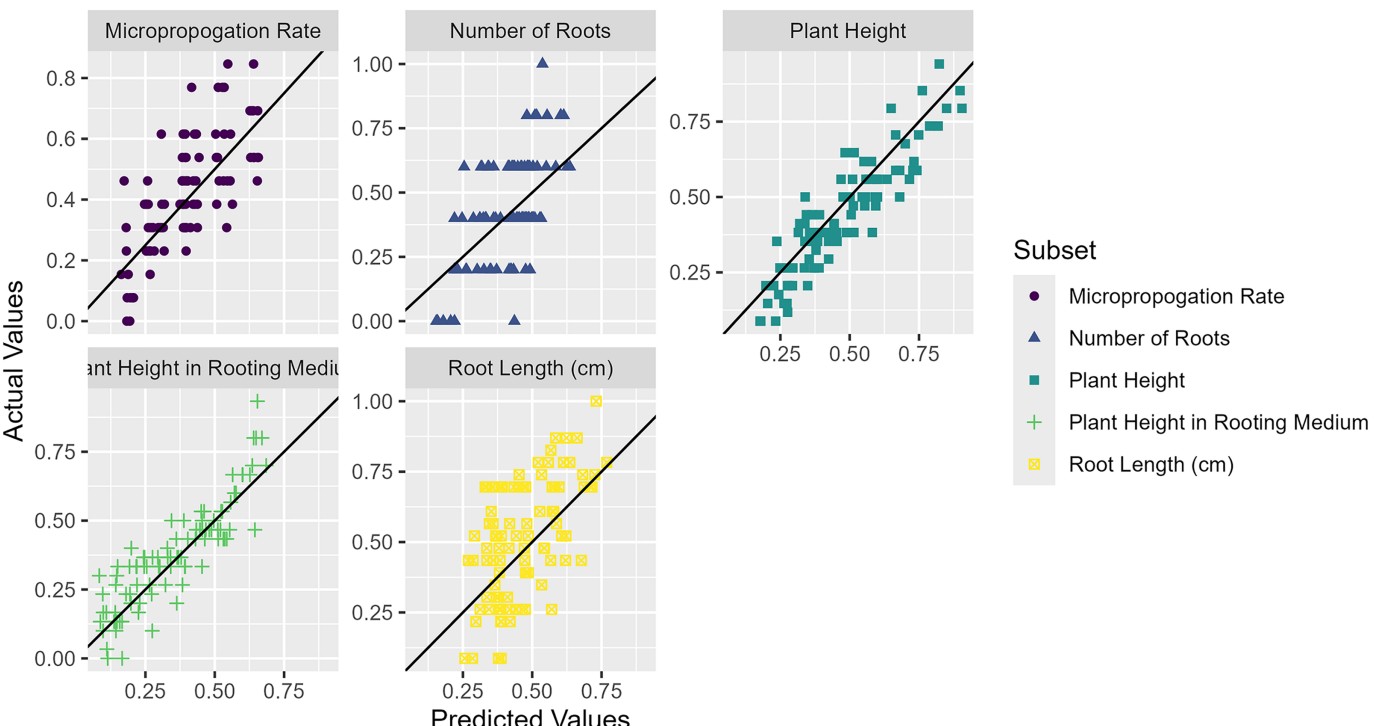

**Figure 5 Scatter plot actual against predicted values using MLP model analysis.**

## DISCUSSION

In discussing our findings on the impact of PEG-induced stress in myrtle, it is pertinent to contextualize our results within the spectrum of existing research. This approach highlights the unique contributions of our study and integrates it into the broader scientific discourse on myrtle's response to various stressors. Our study's observation of decreased growth rates and developmental parameters in myrtle under increasing concentrations of PEG aligns with the findings of *Tafreshi et al. (2021)*. They reported a reduction in shoot regeneration and physiological parameters under similar stress conditions, thereby underlining myrtle's sensitivity to water stress. The novelty of our research lies in its detailed quantification of the specific impact on different genotypes of myrtle, which enriches the understanding of genotype-specific responses to stress. The relation of our findings to the work of *Di Cori et al. (2013)* and *Aghaie, Hosseini Tafreshi & Toghyani (2022)* on salt stress is also notable. Both drought stress (induced by PEG in our study) and salt stress affect the osmotic potential around plant roots. The decrease in growth and development we observed under PEG-induced stress parallels the growth reduction seen under salt stress, suggesting a potentially shared osmotic stress response mechanism in myrtle. Intriguingly, our results contrast with those of *Azizi et al. (2021)*, who demonstrated that microbial inoculation could mitigate the effects of water deficit in myrtle. While we observed a clear decline in various growth parameters with increasing PEG concentrations, *Azizi et al.*'s *(2021)* findings indicate that certain treatments could ameliorate these negative impacts. This disparity underscores the complexity of myrtle's response to environmental stress and suggests the potential for developing strategies to enhance myrtle's resilience. Furthermore, the integration of our findings with propagation studies such as those by *Şimşek et al. (2023)*, *Aka Kaçar et al. (2020*, *2015)*, and *Şan, Karakurt & Dönmez (2015)* is essential. These studies, focusing on myrtle propagation, underline the importance of optimizing growth conditions. The challenges posed by environmental stressors like PEG in tissue culture and propagation efforts highlighted in our study add a critical dimension to this narrative, specifically illustrating how PEG can adversely affect these processes.

In the response of myrtle to PEG-induced drought stress, it is essential to consider the broader context of how drought stress impacts various plant species, particularly in fruit development. This comprehensive understanding aids in drawing parallels and distinctions between our results and those observed in other plants, thereby enriching the insights gained from our study. The research conducted by *Zekai et al. (2022)* on banana cultivars under *in vitro* drought stress provides a vital molecular perspective. They observed differential expression levels in the CDPK gene family, integral to drought stress response, among different banana cultivars. This cultivar-specific molecular response is particularly instructive for our study on myrtle. Just as banana cultivars exhibited varied gene expression responses to drought, the different myrtle genotypes in our study may also have distinct molecular and physiological responses to PEG-induced stress. Understanding these responses at a genetic level can be crucial for developing drought-resilient myrtle

cultivars. Furthermore, studies on various fruits, such as those by *Castellarin et al. (2007)* on grapevine, *Jiang et al. (2020)* on jujube, and *Rathnayaka et al. (2021)* on chili pepper, have shown that drought stress significantly alters biochemical compositions. These alterations include changes in flavonoids, anthocyanins, sugars, and capsaicinoids, all of which are crucial for fruit quality and nutritional value. Our study on myrtle aligns with these findings, suggesting that PEG-induced drought stress similarly impacts metabolic pathways in myrtle, affecting its growth and development. The work of *Asakura et al. (2021)* on tomatoes further highlights the diverse impacts of drought stress, demonstrating an increase in fatty acid synthesis and alterations in fruit yield and ripening periods. These changes are reflective of the complex ways in which plants respond biochemically and physiologically to water scarcity. Incorporating these insights into our study, it becomes evident that drought stress, whether natural or simulated, as in the case of PEG, can have profound effects on plant physiology and biochemistry. This comparison with other species underlines the importance of examining such stress impacts in both agricultural and ornamental plants like myrtle. It also suggests that myrtle might exhibit changes in key metabolic pathways under drought conditions similar to other species, which could have implications for its use and cultivation. *Devireddy et al. (2021)* emphasize the importance of understanding the integration of reactive oxygen species and hormone signaling during abiotic stress. This aspect is also relevant to our study, as these underlying biochemical and molecular mechanisms may also be at play in myrtle's response to drought stress, influencing its growth and development. Our study on myrtle underscores the complexity of plant responses to drought stress when viewed in conjunction with these broader findings. This comparative analysis deepens our understanding of how different species, including myrtle, react to such stressors and opens up new avenues for research and strategies to mitigate drought stress impacts. Developing a comprehensive approach that considers both genetic and biochemical responses could lead to cultivating plant varieties that are more resilient to drought, enhancing both agricultural productivity and the preservation of ornamental species. In conclusion, our research contributes significantly to the current understanding of myrtle's response to drought stress, particularly that induced by PEG. By echoing the findings of reduced growth under stress seen in other studies and providing new insights into genotype-specific responses, our study not only situates itself within the larger body of myrtle research but also opens pathways for future investigations, particularly in the realm of developing strategies to bolster myrtle's resilience against environmental stressors.

In response to drought stress, plants initiate a range of physiological and molecular adaptations to maintain cellular integrity and homeostasis. Our study's findings on the variation in drought tolerance among myrtle genotypes correlate with the observed differential regulation of key osmoprotective compounds and antioxidant enzymes. This observation aligns with these molecules' roles in mitigating the detrimental effects of osmotic stress and oxidative damage (*Şimşek et al., 2024b*). Furthermore, the increased expression of stress-related genes, as seen in our experimental outcomes, mirrors the molecular adaptations described by *Tafreshi et al. (2021)*, who highlighted the upregulation

of drought-responsive genes under controlled stress conditions induced by PEG treatments. These findings underscore the importance of a multi-faceted approach to understanding plant drought tolerance, integrating physiological and genetic insights. We can better understand how plants manage water deficit by elucidating the specific physiological pathways involved in drought response, as observed in our study and supported by external literature. This comprehensive perspective is vital for developing strategies to enhance crop resilience in the face of increasing drought incidence due to climate change.

The research presented in our study and the cited literature collectively highlight the growing integration of ML algorithms in plant tissue culture and their significant impact on enhancing the efficiency and predictability of various *in vitro* processes. This comparative discussion aims to explore the utilization of ML in plant tissue culture, focusing on different studies and their unique contributions to the field.

Our study utilized a comprehensive approach involving tissue culture, rooting trials, and acclimatization processes, emphasizing the genotype's responses to drought stress. We observed that different genotypes exhibited varying micropropagation rates under increasing concentrations of PEG, with Black Fruited genotypes showing higher rates than White Fruited ones. This aspect of genotype-specific responses is echoed in *Rezaei et al. (2023)*, where the unpredictable and genotype-dependent callogenesis in petunia was addressed using ML. Both studies underscore the importance of genotype selection and optimization in tissue culture, which ML algorithms can efficiently handle. The use of different ML algorithms (MLP, GP, XGBoost, SVM, RF) in our study to model various plant growth parameters showcases the versatility of ML in handling diverse aspects of plant tissue culture. This versatility is similarly demonstrated in *García-Pérez et al. (2020)*, where artificial neural networks were applied to optimize the extraction of phenolic compounds in *Bryophyllum* species. The diverse ML applications in both studies signify its potential in a wide array of tissue culture processes. *Kirtis, Aasim & Katırcı (2022)* and *Aasim et al. (2022b)* both emphasize the role of ML in optimizing *in vitro* regeneration protocols, a theme that aligns well with our study. The use of ML to predict outcomes in tissue culture, as seen in the successful regeneration of desi chickpea and common bean, parallels our findings, where ML models efficiently predicted the micropropagation rates and other growth parameters. *Aasim et al. (2022b)* and *Jafari & Daneshvar (2023)* further expand on the utility of ML in plant tissue culture, focusing on hemp and *Passiflora caerulea*, respectively. Both studies highlight the role of ML in optimizing germination and regeneration protocols, resonating with our study's emphasis on the predictive power of ML in tissue culture. *Şimşek et al. (2024a)* and *Jafari & Shahsavar (2020)* demonstrate the application of ML in enhancing the efficiency of micropropagation and rooting protocols in lavender and modeling morphological responses of citrus to drought stress, respectively. These studies, along with our own, illustrate the broad applicability of ML in various aspects of plant tissue culture, from micropropagation to stress response modeling. In conclusion, these studies' collective insights reveal that ML is a powerful tool in plant tissue culture, offering enhanced predictability, optimization, and efficiency in various processes.

The ability of ML algorithms to handle complex datasets and provide precise predictions is invaluable, particularly in genotype-specific responses and optimization of growth conditions. The integration of ML in plant tissue culture research represents a significant advancement in the field, paving the way for more controlled, efficient, and productive cultivation practices.

Our study provides a foundation for further research into the response of myrtle genotypes to drought stress, but several areas remain ripe for exploration. Future studies could expand the scope by investigating the effects of other abiotic stressors, such as salinity or temperature extremes, on myrtle's physiological and genetic responses. Integrating advanced ML models, such as deep learning or ensemble techniques, could enhance our analyses' accuracy and predictive power. Exploring the molecular mechanisms underlying the observed genotype-specific responses, perhaps through genomic or transcriptomic studies, could offer deeper insights into the resilience traits of myrtle. Furthermore, comparative studies across different plant species could help identify universal stress response mechanisms and contribute to broader agricultural and ecological applications. Such interdisciplinary approaches will broaden our understanding of myrtle's adaptive capabilities and aid in developing robust strategies for cultivating drought-resistant plant varieties in the face of climate change.

## CONCLUSION

The investigation into the performance of superior myrtle genotypes under *in vitro* drought conditions has provided valuable insights into the adaptive capabilities of these plants. Our study has shown that myrtle genotypes respond differently to increasing concentrations of polyethylene glycol, a surrogate for drought stress. Particularly, Black Fruited genotypes exhibited higher resilience in terms of micropropagation rates compared to their White Fruited counterparts. The application of machine learning models has been instrumental in predicting and analyzing the performance of these genotypes under stress. The models demonstrated a high degree of accuracy, indicating their potential as reliable tools in the field of plant tissue culture and stress response analysis. This approach enhances our understanding of genotype-specific responses to environmental stresses and paves the way for more efficient and targeted cultivation strategies. Overall, our findings underscore the importance of considering genotype-specific traits in developing cultivation practices for myrtle, especially in the face of increasing drought conditions due to climate change. The successful integration of machine learning into this field offers a promising direction for future research and application, potentially revolutionizing plant tissue culture and breeding strategies for enhanced resilience against environmental stresses.

## ACKNOWLEDGEMENTS

The authors thank Eugene Steele, a professional English editor, for editing the manuscript in English.

### Funding

This research was funded by Erciyes University Scientific Research Projects Units, grant number FYL-2023-12821. The Office of the Dean for Research at Erciyes University also provided the necessary infrastructure and laboratory facilities at the ArGePark research building. The funders had no role in study design, data collection and analysis, decision to publish, or preparation of the manuscript.

### Grant Disclosures

The following grant information was disclosed by the authors:
Erciyes University Scientific Research Projects Units: FYL-2023-12821.
The Office of the Dean for Research at Erciyes University.

### Competing Interests

Taner Bozkurt is employed by Tekfen Agricultural Research Production and Marketing Inc.

### Author Contributions

- Ümit Bektaş performed the experiments, prepared figures and/or tables, and approved the final draft.
- Musab A. Isak performed the experiments, prepared figures and/or tables, and approved the final draft.
- Taner Bozkurt performed the experiments, authored or reviewed drafts of the article, and approved the final draft.
- Dicle Dönmez performed the experiments, prepared figures and/or tables, and approved the final draft.
- Tolga İzgü analyzed the data, authored or reviewed drafts of the article, and approved the final draft.
- Mehmet Tütüncü analyzed the data, authored or reviewed drafts of the article, and approved the final draft.
- Özhan Simsek conceived and designed the experiments, analyzed the data, prepared figures and/or tables, authored or reviewed drafts of the article, and approved the final draft.

### Data Availability

The raw data is available in the Supplemental File.

### Supplemental Information

Supplemental information for this article can be found online at http://dx.doi.org/10.7717/peerj.18081#supplemental-information.

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
