# Peer review of "Genotype-specific responses to in vitro drought stress in myrtle (Myrtus communis L.): integrating machine learning techniques"

_PeerJ, doi:10.7717/peerj.18081_

## Round 0.1 · original submission · Major Revisions

Reviewers have touched on all the important points. Your manuscript requires major revisions

Reviewer 1 ·

Basic reporting

The manuscript entitled “Genotype-specific responses to in vitro drought-stress in myrtle (Myrtus communis L.): integrating machine learning techniques” provides a detailed account of the experimental findings related to the performance of superior myrtle genotypes under in vitro drought conditions has provided valuable insights into the adaptive capabilities of these plants. However, addressing the below-mentioned points would enhance its clarity, specificity, and impact.
1. The abstract provides an overview of the experimental findings, it lacks a clear structure that guides the reader through the different aspects of the study. Organizing the abstract into distinct sections (e.g., background, methods, results, conclusions) would improve clarity and facilitate understanding.
2. The abstract should conclude with a succinct summary of the main findings and their broader implications. It would be beneficial to provide interpretations or hypotheses in the abstract regarding the implications of these changes on stress tolerance mechanisms. Ensure the concluding sentences highlight the practical implications of your findings for cultivation practices under climate change.

Experimental design

3. The introduction section lacks research questions and a hypothesis. Please formulate a hypothesis. Keep in mind that the research objectives should stem from the research hypothesis. Additionally, ensure that the objectives are stated clearly.
4. In the methodology section, it is mentioned that the selection of the genotypes is vaguely described. Critical visual parameters are mentioned, but there is no mention of specific quantitative criteria or the number of plants initially screened. Provide more detailed information on the selection criteria.
5. The sterilization steps are clearly outlined, employing 70% ethanol and 20% sodium hypochlorite for sterilization. However, it raises a question as to why mercuric chloride, a commonly used and highly effective sterilizing agent, was not included in your protocol. The exclusion of mercuric chloride could potentially result in significant contamination issues. Could you please elaborate on the rationale behind not selecting mercuric chloride for your study?
6. The concentrations of PEG6000 used (1%, 2%, 4%, and 6%) might be too low to effectively simulate a wide range of drought conditions. Typically, higher concentrations are needed to induce more severe drought stress. Methodology does not provide a rationale for selecting these specific concentrations. There is no reference to previous studies or preliminary data that justify why these concentrations are appropriate for inducing drought stress in myrtle. The current range may not cover severe drought stress adequately, which could limit the understanding of the upper limits of myrtle genotypes' drought tolerance. While implied, the methodology does not explicitly mention a control (0% PEG6000) for baseline comparison. Based on standard practices in crop physiology, here is a suggested range of PEG6000 concentrations:
Low stress: 5-10% PEG6000 (For mild drought stress and initial screening).
Moderate stress: 15-20% PEG6000 (To differentiate between drought-tolerant and sensitive genotypes).
High stress: 25-30% PEG6000 (To simulate severe drought conditions and study the extreme responses).

Validity of the findings

The authors have presented the results of several studies, which are intriguing; however, these findings are not clearly linked. It is challenging to discern the overarching purpose of the study and the significance of the results obtained. Moreover, the paper lacks well-defined conclusions.

While the results present quantitative data, they could benefit from more in-depth interpretation and discussion of the observed trends. For instance, the underlying physiological mechanisms driving the observed changes in stress tolerance are not fully elucidated, limiting the insight into the biological significance of the findings.

Additional comments

Please ensure that all references follow a uniform format throughout the manuscript. Consistency in citation style enhances the readability and professionalism of the manuscript.
Overall, while the results section effectively presents the experimental findings, addressing the above-mentioned points would improve the clarity, completeness, and interpretation of the results.

Reviewer 2 ·

Basic reporting

All comments have been added in detail to the last section.

Experimental design

All comments have been added in detail to the last section.

Validity of the findings

All comments have been added in detail to the last section.

Additional comments

Review Report for PeerJ
(Genotype-specific responses to in vitro drought stress in myrtle (Myrtus communis L.): integrating machine learning techniques)

1. Within the scope of the study, superior myrtle genotypes performance was explored, rooting efficiency and micropropagation were predicted with various machine learning models.

2. In the introduction section, the importance of the subject and the literature were mentioned. In this section, it is suggested to add a literature analysis table. In addition, the importance of the study and its difference from the literature should be stated more clearly in bullet points.

3. When the dataset and data analysis used in the study are examined, it is observed that they are at a sufficient level.

4. When the results obtained in the study and the evaluation metric types are examined in terms of the analysis of the study, it is understood that they are at an acceptable level.

5. Many different models such as XGBoost. random forest, support vector machine etc. were preferred in the study as machine learning models. The variety of models used and the reasons for preference are very important for the correct analysis of the results. Although there are many different machine learning models that can be used for problem solving within the scope of this study, it should be stated more clearly why these models were preferred and/or whether different experiments were conducted.

6. Although the discussion and conclusion sections seem to be sufficient in principle, it is recommended that detailed information be provided regarding the future works section in order to increase the contribution of the study to the literature and its potential to be cited after publication.

In conclusion, although the study is important in terms of the problem addressed, it should definitely be examined in terms of the sections listed above.

---

## Round 0.2 · accepted · Accept

You have improved the manuscript by making the necessary changes for it to be accepted. Congratulations.

Reviewer 1 ·

Basic reporting

Satisfactory. Self-contained with relevant results to hypotheses.

Experimental design

Research question well defined, relevant & meaningful. It is stated how research fills an identified knowledge gap.

Validity of the findings

All underlying data have been provided; they are robust, statistically sound, & controlled.

Reviewer 2 ·

Basic reporting

All comments have been added in detail to the last section.

Experimental design

All comments have been added in detail to the last section.

Validity of the findings

All comments have been added in detail to the last section.

Additional comments

Review Report for PeerJ
(Genotype-specific responses to in vitro drought stress in myrtle (Myrtus communis L.): integrating machine learning techniques)

Thank you for the revision. The new version of the paper and the responses to the reviewer comments have been examined in detail. Although some responses are limited, the overall responses and relevant changes are sufficient. Therefore, I recommend that the paper be accepted. I wish the authors success in their future papers and projects. Best regards.